# Comparison between Gibson–Cooke and Macroduct Methods in the Cystic Fibrosis Neonatal Screening Program and in Subjects Who Are Cystic Fibrosis Screen-Positive with an Inconclusive Diagnosis

**DOI:** 10.3390/ijns9030041

**Published:** 2023-07-25

**Authors:** Daniela Dolce, Cristina Fevola, Erica Camera, Tommaso Orioli, Ersilia Lucenteforte, Marco Andrea Malanima, Giovanni Taccetti, Vito Terlizzi

**Affiliations:** 1Cystic Fibrosis Regional Reference Center, Department of Paediatric Medicine, Meyer Children’s Hospital IRCCS, Viale Gaetano Pieraccini 24, 50139 Florence, Italy; cristina.fevola@meyer.it (C.F.); erica.camera@meyer.it (E.C.); tommaso.orioli@meyer.it (T.O.); giovanni.taccetti@meyer.it (G.T.); 2Unit of Medical Statistics, Department of Clinical and Experimental Medicine, University of Pisa, 56126 Pisa, Italy; ersilia.lucenteforte@unipi.it (E.L.); marco.malanima@gmail.com (M.A.M.)

**Keywords:** sweat test, Gibson–Cooke method, macroduct system-based method, CRMS/CFSPID, QNS, cystic fibrosis outcome

## Abstract

The sweat test (ST) is the current diagnostic gold standard for cystic fibrosis (CF). Many CF centres have switched from the Gibson–Cooke method to the Macroduct system-based method. We used these methods simultaneously to compare CF screening outcomes. STs using both methods were performed simultaneously between March and December 2022 at CF Centre in Florence. We included newborns who underwent newborn bloodspot screening (NBS), newborns undergoing transfusion immediately after birth, and children with CF screen-positive, inconclusive diagnosis (CFSPID). We assessed 72 subjects (median age 4.4 months; range 0–76.7): 30 (41.7%) NBS-positive, 18 (25.0%) newborns who underwent transfusion, and 24 (33.3%) children with CFSPID. No significant differences were found between valid sample numbers, by patient ages and groups (*p* = 0.10) and between chloride concentrations (*p* = 0.13), except for sweat chloride (SC) measured by the Gibson–Cooke and Macroduct methods in CFSPID group (29.0, IQR: 20.0–48.0 and 22.5, IQR: 15.5–30.8, respectively; *p* = 0.01). The Macroduct and Gibson–Cooke methods showed substantial agreement with the SC values, except for CFSPID, whose result may depend on the method of sweat collection. In case of invalid values with Macroduct, the test should be repeated with Gibson–Cooke method.

## 1. Introduction

Cystic fibrosis (CF) is an autosomal recessive genetic disease caused by mutations in a single gene located on the long arm of chromosome 7. CF is a chronic disease characterised by suppurative lung disease, pancreatic insufficiency, multifocal biliary cirrhosis, male infertility, and excessive loss of salts through sweat [1].

A diagnosis of CF is confirmed when sweat chloride (SC) values in two independent measurements are >60 mmol/L or when two disease-causing variants of CF are found in the sequence of the CF transmembrane conductance regulator (*CFTR*) gene [2]. SC values are differentiated by age: for infants aged < 6 months, values < 30 mmol/L are considered normal, and those between 30 mmol/L and 59 mmol/L are considered to be in the intermediate range; for babies aged ≥ 6 months, values < 40 mmol/L are considered normal, and values between 40 and 59 mmol/L are considered to be in the intermediate range [3,4,5,6].

Although several methods have been proposed for collecting and measuring electrolytes from sweat, the method described by Gibson and Cooke [7] in 1959 is a reference method recommended by national and international guidelines [6,8,9].

The Gibson–Cooke method is based on the detection of elevated SC values using the quantitative pilocarpine iontophoresis test (QPIT). The QPIT sweat test (ST), as originally described, is remarkably accurate, but it requires careful application of the collecting pad to prevent sweat evaporation, the use of a chemical balance, and calculation of concentrations from eluted specimens containing widely varying quantities of sweat [10]. To perform an accurate ST, at least 75 mg of sweat must be collected [11,12]. Nevertheless, this can be difficult, especially in infants aged < 3 months, as already reported by Beauchamp et al. in a Canadian multicenter study where not sufficient quantity (QNS) of sweat volume resulted in 18.3% of tests [13].

To simplify the test, many clinical centres and laboratories use alternative methods [12,14], such as the Macroduct^®^ Advanced Sweat Collection System (Wescor Inc., Logan, UT, USA). This system safely and effectively stimulates human sweat via iontophoresis using 0.5% pilocarpine gel discs, with subsequent sweat collection in a capillary. By using this closed system for sweat collection and the measurement of sweat weight, the risk of evaporation is eliminated. Additionally, a small amount of dye on the collection surface facilitates the visualisation of the stored sweat, allowing it to be quantified in microliters [10].

A comparison using the two collection methods was already carried out by Mastella et al. in a prospective study on 318 subjects with Macroduct and on 305 with Gibson–Cooke method. An adequate amount of sweat was in 90.9% and 96.4% of collections, respectively, and sensitivity and specificity were comparable [15]. The correlation of the two methods regarding the chloride values was also evaluated by Rose et al., with a discrepancy in the results in 22% of 82 subjects (3.7–60.1 years) [16]. 

However, no studies have compared QNS rates and SC values using the two collection methods in newborn bloodspot screening (NBS)-positive subjects or those with *CFTR*-related metabolic syndrome/CF screen-positive, inconclusive diagnosis (CRMS/CFSPID). The latter group comprised NBS-positive infants with an inconclusive CF diagnostic test result, having ST results in the intermediate range and/or <2 CF-causing variants and normal SC [17,18]. Because most subjects with CFSPID are asymptomatic, the ST is the only routinely available test to make a definitive CF diagnosis or to reach an early diagnosis in a healthy subject or carrier [18,19,20,21]. The prevalence of CFSPID is highly variable across different countries and depends on the screening algorithm used. No study has evaluated whether the method used for sweat collection can influence the SC value and, therefore, the final diagnosis of these patients. Diagnostic tests for CF can cause anxiety in parents because of the waiting time and uncomfortableness involved [22,23].

The primary objective of the present study was to compare the results of the Macroduct system-based method and the classical Gibson–Cooke method in a pediatric cohort. In particular, we evaluated the following: (1) percentage of valid sweat samples in infants aged < 1 year who were CF NBS positive or subjects who had undergone transfusion in the first 72 h after birth; (2) percentage of valid sweat samples in children with CFSPID aged 1–6 years; and (3) amount of chloride for each test measured using the two methods.

## 2. Materials and Methods

### 2.1. Study Population

In this prospective, non-randomized, single-centre study, the ST was performed simultaneously using the Gibson–Cooke and Macroduct system-based method between March and December 2022. The inclusion criteria were as follows: (1) newborns who tested positive during the NBS program, according to the algorithm blood immunoreactive trypsinogen (b-IRT)-meconium lactase-DNA-ST [24]; (2) early transfused newborns undergoing red blood cell or platelet transfusion in the first 72 h after birth, in whom the b-IRT levels were not measured during NBS for CF; and (3) children with CFSPID, according to the criteria described by Ren [17], in whom the ST was performed to reach a definitive diagnosis.

For all newborns, the levels of b-IRT were measured from a blood spot sample taken on the third day of life using the GSP instrument (Perkin-Elmer, Turku, Finland). The b-IRT cut-off value ≥99th percentile, calculated every 4 months, was 47–50 ng/mL. All newborns with a b-IRT value ≥99th percentile underwent *CFTR* genetic analysis, including all *CFTR*-causing variants, according to the *CFTR*2 database “https://cftr2.org/ (accessed on 5 April 2023)”. For transfused newborns, we usually perform ST directly to exclude CF, as NBS results could be unreliable due to low weight or stress during childbirth [25,26].

All children with CFSPID or infants with one *CFTR* variant at the first level underwent more extensive DNA testing (*CFTR* gene sequencing and multiplex ligation-dependent probe amplification to identify large deletions/insertions).

This study was approved by the regional ethical committee for pediatric clinical trials (Florence, Ethics Clearance number 317/2021, on 14 December 2021). Informed consent was obtained from all the parents (or legal guardians) to perform the two diagnostic tests and obtain anonymous clinical data for research purposes.

### 2.2. Sweat Test

All the STs were performed by experienced nurses who performed at least 200 diagnostic STs annually from the stimulation to SC collection using the Gibson–Cooke method. For the Gibson–Cooke method, sweat secretion was stimulated by iontophoresis for 5 min (total applied current of 1.5 mA, 50 μA/cm^2^) using 0.5% pilocarpine gel discs [7]. The preferred site for the ST stimulation was the lower portion of the forearm flexor; however, the inner thigh was used if the entire arm was too small to attach the collector (as in the case of a premature infant). After cleansing the skin with distilled water, sweat was collected by placing a Whatman 541-type filter paper with an area of approximately 25 cm^2^, covered and sealed with polyethylene film, over the stimulated area [6,7,8]. Sweat samples were collected over 30 min. The Macroduct system was used simultaneously on the other forearm or thigh, and sweat secretion was stimulated using the same method. After cleaning the stimulated area with distilled water, sweat was collected in a Macroduct coiled plastic tubing collector cup for up to 30 min [10]. The SC concentrations of samples collected using both methods were determined by dedicated laboratory personnel using a chloride analyser (MKII Chloride Analyzer 926S, Sherwood Scientific Ltd., Cambridge, UK). Sweat collections were considered insufficient if <75 mg of samples were collected using the Gibson–Cooke method and <15 μL using the Macroduct method [6,8,9]. Additionally, the CF Centre in Florence participates in the External Quality Assessment Program for the ST of the Istituto Superiore di Sanità (ISS, Rome, Italy) [27,28].

### 2.3. Statistical Analyses

Categorical variables were reported as numbers and percentages, and continuous variables were reported as means and standard deviations or medians and interquartile ranges [IQRs], as appropriate. The Shapiro–Wilk test was used to assess the normality of the continuous data.

We analysed the differences in valid test numbers and SC concentration values for the Gibson–Cooke and Macroduct methods. McNemar’s test with continuity correction was used to assess the differences in categorical variables. To assess the differences between continuous normally distributed variables, paired Student’s *t*-tests were used, whereas for continuous non-normally distributed variables, paired Wilcoxon tests were used. Differences were considered statistically significant if *p*-values were less than 0.05. Finally, we estimated the diagnostic agreement between the two methods by calculating Cohen’s kappa coefficient and corresponding 95% confidence intervals (CIs). Cohen’s interpretation of the value of kappa was performed according to Landis and Koch [29]. Statistical analyses were performed using R Statistical Software (v4.2.2; R Core Team 2022).

## 3. Results

Simultaneous Gibson–Cooke and Macroduct tests were conducted on 72 subjects (32 males), with a median age of 4.4 months [IQR: 1.8–18.8]. The participants belonged to the following groups: infants who were NBS-positive, 30 (41.7%); newborns who underwent transfusion, 18 (25.0%); and children labelled as CFSPID, 24 (33.3%) (Table 1).

No statistically significant differences were observed between the percentages of valid tests with the Gibson–Cooke and Macroduct methods (Table 2). We divided the results by both subject type (NBS-positive, transfused newborns and CFSPID) and age group. In addition, we examined whether the weight in the two groups in NBS-positive and transfused infants affected the number of invalid tests, showing no statistical difference either with the Gibson–Cooke and/or Macroduct methods (*p* = 0.40 and *p* = 0.29, respectively). 

Differences in SC levels were not statistically significant between the two methods, with a median of 16 mmol/L [IQR: 9.3–29.8] and 16 mmol/L [IQR: 11.0–23.0] for the Gibson–Cooke and Macroduct system-based method, respectively (*p* = 0.13). However, there was a significant difference between the mean values of SC measured by the Gibson–Cooke and Macroduct system-based method in the children with CFSPID [29.0, IQR: 20.0–48.0 and 22.5, IQR: 15.5–30.8 respectively] (*p* = 0.01; Figure 1).

The two methods were in substantial agreement for the number of valid tests, as reported by the diagnostic category (Table 3). Cohen’s kappa coefficient for the NBS group was 0.93 (95% CI: 0.79–1.00); for the CSFPID group, it was 0.53 (0.23–0.83), while in the group of newborns who underwent transfusion, the agreement was total.

Five children with CFSPID and one screen-positive child showed discordant results. The genetic information of the participants with discrepant diagnoses between the two tests is summarized in Table 4.

## 4. Discussion

In this study, we compared two SC collection methods, the Macroduct system and the classical Gibson–Cooke method. Although other papers have already compared the two methods, also on larger cohorts, there is no comparison data on positive CF-NBS infants or children labelled as CFSPID. 

We observed no significant differences between the number of valid samples with respect to patient ages and groups. However, a higher percentage of valid tests was noted with Gibson–Cooke and this difference may have more impact during a large-scale screening of newborns. 

Regarding the diagnostic category, the two methods were in substantial agreement in all patient groups except for children with an inconclusive diagnosis. In this sub-group, the mean SC value measured using the Gibson–Cooke method was significantly higher than that measured using the Macroduct system-based method. This affected the diagnostic category of five (20.8%) out of 24 children with CFSPID: for four children, from an intermediate value to a normal value, and for one child, from a pathological value to an intermediate value. Our data showed how the sweat collection method affected chloride values in children with CFSPID and, therefore, the definitive diagnostic category of a child with a still inconclusive diagnosis, thereby making CFSPID outcomes non-comparable in different cohorts. These data are important because all cases of CFSPID evolved into cases of CF during the follow-up at our centre, and most of the cases described in other cohorts involved asymptomatic children with pathological SC results [18,30,31]. Therefore, defining the method of sweat collection or a possible change to other methods can modify CF diagnosis in patients with CFSPID, causing parental distress, anxiety, and uncertainty for families [23,32]. In one case, there was a non-concordant result for the interpretative category among the NBS-positive patients: the SC value varied from 31 mmol/L to 27 mmol/L with the two methods, thereby changing the result from an intermediate to a negative value.

The UK guidelines for ST require laboratories to maintain an annual QNS of no more than 10% of the population tested. The goal should be to have a QNS rate of less than 5% in children over 6 months of age. In children under 6 months of age, failed sweat collections should not exceed 20% of the tested population [8]. This requirement prevents repeated testing triggered by insufficient sample volume, which, in turn, increases the waiting time for a definitive diagnosis, delays the initiation of therapy, and reduces the overall cost-effectiveness of the ST. However, obtaining a sufficient sample volume remains a major challenge for SC testing [33,34].

We observed a slightly higher failure rate using the Macroduct system-based method. This could be due to several factors, including the smaller sweat collection surface area with respect to the filter paper method and the elaborate process of recovering sweat from the Macroduct’s small coiled plastic tube with a syringe, as reported by Rose [16]. However, this method had several advantages since it allowed for the direct analysis of SC values without the need to use an analytical balance, dilute the sample and calculate the result after considering the dilution factor. However, re-collection using Gibson–Cooke method may be necessary for those failed by Macroduct. Furthermore, as shown in Table 4, the lower bias of SC by Macroduct leads to the possibility of the lowest chloride values in CF children previously CFSPID. A method-dependent cutoff of SC collection should be evaluated on a larger cohort of positive NBS infants. 

Our study has several limitations. The comparison was carried out on a small number of participants, and the three groups were not comparable. Furthermore, we could not perform the tests in a control group (i.e., among the participants who tested CF-negative during NBS). A period of nursing training would have been helpful prior to enrolling the participants and obtaining more data on the same participants by repeating the tests over time. However, we provided prospective data comparing the two sweat collection methods on a previously unstudied cohort, thereby demonstrating the validity of the Macroduct system-based method and highlighting the limitations of sweat testing as a diagnostic test for children with CFSPID.

## 5. Conclusions

Sweat chloride collection with the Macroduct system in CF NBS-positive infants or in those who underwent transfusion or were labelled as CFSPID may be associated with a higher percentage of invalid tests. A repeated collection using Gibson–Cooke method may be necessary for those failed by Macroduct. Furthermore, lower SC values with the Macroduct system could influence the detection and outcomes of CFSPID children.

## Figures and Tables

**Figure 1 IJNS-09-00041-f001:**
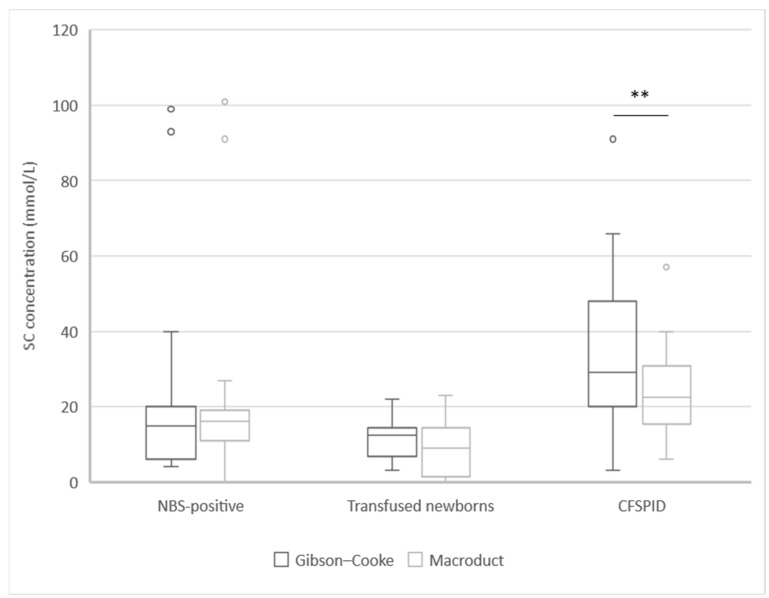
Sweat chloride concentration analysed with the Gibson–Cook method and the Macroduct system in three groups of subjects. ** Statistically significance *p* ≤ 0.01. Abbreviations: CFSPID: CF screen-positive, inconclusive diagnosis; NBS: newborn bloodspot screening; SC: sweat chloride.

**Table 1 IJNS-09-00041-t001:** Characteristics of study participants.

Type of Participants	Number of Patients (%)	Males/Females (*n*)	Age in Months, Median (IQR)
NBS-positive *	30 (41.7)	15/15	1.9 (1.4–3.1)
Transfused newborns **	18 (25.0)	11/7	3.6 (2.0–7.5)
CFSPID	24 (33.3)	7/17	25.0 (18.6–50.5)

* Two patients were diagnosed with CF: one male aged 70 days and one female aged 28 days; 10 healthy carriers; 15 healthy subjects. ** All transfused newborns were negative for sweat tests. Abbreviations: CFSPID: cystic fibrosis screen-positive, inconclusive diagnosis; IQR: interquartile range; NBS: newborn bloodspot screening; ST: sweat test.

**Table 2 IJNS-09-00041-t002:** Number of valid tests with the Gibson–Cooke and Macroduct methods in three participant groups.

	Number of Patients (%)	Valid Test Gibson–Cooke Method (%)	Valid Test Macroduct Method (%)	*p*-Value
Strata according to type of patients				
NBS-positive	30 (41.7)	27 (90.0)	25 (83.3)	0.68
Transfused newborns	18 (25.0)	14 (77.8)	13 (66.7)	0.62
CFSPID	24 (33.3)	23 (95.8)	20 (83.3)	0.25
Strata according to age of patients				
Months				
<12	50 (69.4)	43 (86.0)	39 (78.0)	0.34
12	8 (11.1)	8 (100)	6 (75.0)	0.48
24	6 (8.3)	6 (100)	6 (100)	NE
36	2 (2.8)	2 (100)	2 (100)	NE
48	1 (1.4)	-	-	NE
60	4 (5.6)	4 (100)	3 (75.0)	1.00
72	1 (1.4)	1 (100)	1 (100)	NE

Abbreviations: CFSPID: CF screen-positive, inconclusive diagnosis; NE: not estimable; NBS: newborn bloodspot screening.

**Table 3 IJNS-09-00041-t003:** Comparison of sweat chloride results using the Gibson–Cooke and Macroduct methods.

		Normal	Intermediate	Pathologic	QNS
NBS-positive	Gibson–Cooke	23	2	2	3
	Macroduct	23	0	2	5
Transfused newborns	Gibson–Cooke	14	0	0	4
	Macroduct	13	0	0	5
CFSPID	Gibson–Cooke	14	7	2 *	1
	Macroduct	17	3	0	4

* With Macroduct method 1 = QNS and 1 = in intermediate range. Abbreviations: CFSPID: CF screen-positive, inconclusive diagnosis; NBS: newborn bloodspot screening; QNS: quantity not sufficient.

**Table 4 IJNS-09-00041-t004:** Genotype, sweat chloride values, and diagnosis/label at the end of the observation period for children with whom the results of the two methods did not agree. Final diagnoses were assessed using the results obtained with the Gibson–Cooke method.

Patients Number	Patient’s Categorization *	Sex	Age (Months)	Gibson–Cooke ([Cl^−^] mmol/L)	Macroduct ([Cl^−^] mmol/L)	*CFTR* Genotype *	Final Diagnosis/ Label
12	CFSPID	M	9.0	50	30	F508del/S737F	CFSPID
13	CFSPID	F	66.6	49	33	G542X/UN	CFSPID
18	CFSPID	M	77.9	48	17	F508del/D1152H	CF ˆ
21	CFSPID	F	25.7	66	57	F508del/S737F	CFSPID
43	CFSPID	F	69.9	41	22	Dele2ins182/5T;TG11	CFSPID
71	Screening positive	F	3.8	31	27	F508del/UN	CF carrier

ˆ CF diagnosis for pathological sweat test, in absence of symptoms. Abbreviations: CFSPID: cystic fibrosis screen-positive, inconclusive diagnosis: CF: Cystic Fibrosis; *CFTR*: cystic fibrosis transmembrane conductance regulator; UN: unknown. * After gene sequencing and multiplex ligation-dependent probe amplification.

## Data Availability

The data presented in this study are available on request from the corresponding author. The data are not publicly available due to privacy.

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
