# Peer review of "Comparison between Gibson–Cooke and Macroduct Methods in the Cystic Fibrosis Neonatal Screening Program and in Subjects Who Are Cystic Fibrosis Screen-Positive with an Inconclusive Diagnosis"

_2409-515X, 2023, doi:10.3390/ijns9030041_

Round 1
Reviewer 1 Report
The manuscript is scientifically strong and well-written. However, the study lacks novelty and the results have largely been reported elsewhere. The authors should cite relevant studies (Beauchamp et al., Rose et al., Mastella et al., etc.) in the background section and highlight how the current study builds on previous studies.
The discussion section does not make a strong argument that the data presented are novel and important enough to justify publication. Please highlight the importance of the results more strongly.
The sample size in the current study is very low and not all comparisons were able to be made. Similar studies have had a greater number of participants.
Author Response
Reviewer 1
The manuscript is scientifically strong and well-written. However, the study lacks novelty and the results have largely been reported elsewhere. The authors should cite relevant studies (Beauchamp et al., Rose et al., Mastella et al., etc.) in the background section and highlight how the current study builds on previous studies.
Re: We thank the reviewer for taking the time to review. We have modified the background and added the suggested references. We are aware that other papers have compared the two methods, even on larger cases. The novelty of our study is in group of enrolled subjects, i.e. a pediatric cohort of positive neonatal screening children and above all the comparison of the two methods in CRMS/CFSPID. Furthermore there are no comparison studies on infants transfused before 72 hours of life (a large number of subjects) and therefore directly performing the sweat test.
As written in the introduction “No study has evaluated whether the method used for sweat collection can influence the SC value and therefore the final diagnosis of CRMS/CFSPID”. Our paper shows how the detection and outcome of these subjects (diagnosis/non-diagnosis of CF) can depend on the method used and also making the several cohorts described not comparable.
We believe that this data is relevant, due to the growing number of papers published on CRMS/CFSPID cohorts.
Certainly other results confirm data already reported.
The discussion section does not make a strong argument that the data presented are novel and important enough to justify publication. Please highlight the importance of the results more strongly.
Re: thank you for this suggestion. We strongly believe that our study reports new and useful data for the scientific community. As already written above, the other studies on this topic are on children or adults or subjects with an uncertain diagnosis. We have edited the discussion as suggested.
The sample size in the current study is very low and not all comparisons were able to be made. Similar studies have had a greater number of participants.
Re: We are aware of it. We have reported in the text that other papers on larger cases have already been published. In addition, the low number of enrolled subjects is one of the limits reported in discussion.
Reviewer 2 Report
The authors provide a useful study on the two methods for sweat collection in the context of NBS setting.
1. Table 2, although the statistics are not significant, the exact percentage appeared a consistent lower valid rate for Marcroduct method. This difference may have more impact when taking into large scale screening especially to asymptomatic newborns. Re-collection using Gibson-Cooke method may be necessary for those failed by Macroduct. Together with the lower bias of SC by Marcoduct with the possibility leading to false negative, the authors should mention these limitations in the abstract and the conclusion.
2. Table 4 shows the discordant cases, where SC are consistently lower collected by Macroduct leading to false negative results. Can the author discuss on the possibility of collection method dependent cutoff for CF?
3. It would be helpful to provide a correlation chart for all the results over a wide range of SC comparing the two collection methods.
Author Response
Reviewer 2
The authors provide a useful study on the two methods for sweat collection in the context of NBS setting.
Re: We thank the reviewer for taking the time to review and for the positive comment.
- Table 2, although the statistics are not significant, the exact percentage appeared a consistent lower valid rate for Marcroduct method. This difference may have more impact when taking into large scale screening especially to asymptomatic newborns. Re-collection using Gibson-Cooke method may be necessary for those failed by Macroduct. Together with the lower bias of SC by Marcoduct with the possibility leading to false negative, the authors should mention these limitations in the abstract and the conclusion.
Re: Thank you for your comment. We have revised Table 2 by adding the total number of tests performed next to the percentages. We fully agree with the reviewer and have changed the text in several points as suggested. We are aware that these results may have an impact when looking at higher numbers, leading also to a statistically significant difference between the two groups. We have added this also in discussion.
However, it is important to note that we have not found any false negative results with the use of Macroduct. Patient 4 in table 4 is an asymptomatic CFSPID who had a pathological chloride value with Gibson-Cook and in the intermediate range with Macroduct. In any case, he would have continued the follow up at the CF centre. The other subject (patient 6 of table 4) was a carrier.
We have specified that with Macroduct the lower chloride values can influence both the number of children labelled as CFSPID and the outcome over time of these children. These data are very important especially on a topic like CRMS/CFSPID, given the growing number of papers published.
- Table 4 shows the discordant cases, where SC are consistently lower collected by Macroduct leading to false negative results. Can the author discuss on the possibility of collection method dependent cutoff for CF?
Re: We understand the reviewer's request. Our study is performed on too low number of patients to propose new sweat chloride cutoffs using the Macroduct system versus Gibson-Cook. However, we have mentioned this in the conclusions, hoping for studies on larger case studies.
- It would be helpful to provide a correlation chart for all the results over a wide range of SC comparing the two collection methods.
Re: Thanks for the suggestion, we have added a figure with the SC concentrations comparing the two collection methods in the three groups of subjects.
Reviewer 3 Report
Dear authors, I have read your manuscript with high interest. The direct comparison of these 2 methods could be helpful for centers and clinic to decide which test is more suitable to start with, and also to make a decision to change methods perhaps.
However, there are several points that need minor and sometimes major attention.
Major comments:
1.) The conclusion is merely a repition of results. In addition the authors state, that there is no differnce in the number valid samples. However, there are up to 25% more invalid tests with the macroduct systems. It is somehow unbelievale, that this is not significant.
2.) The discussion should also be completely rewritten. The present text is not very structured, and it includes a lot of redundant information, that is already mentioned in the introduction.
3.) Table 3 is very hard to understand. It would be much better to rearange it, probably in the following way:
normal interm. altered pathologic
Gib.-Cooke 20 1 0
Macroduct 21 0 2
However, as you can see, I also do not understand your numbers. The number Macroduct and Gibson-Cooke do not match, and also the numbers between table 1 and table 2 do not match.
Minor comments:
4.) page 4, lines 157-161: This paragraph is quite unclear written. Please rewrite to make it more concise.
5.) page 4, line 166 - page 5 line 171: The data for SC should be presented in a graph. Additionally, the authors should definitely present the only significant difference. This should be presented in the same graph with different symbols/colours for NBS, CFSPID, transfused
6.) Introduction, page 2 line 58: "The Gibson-Cooke method involves a significant risk of chloride evaporation ... "
First of all chloride from an aqueous solution cannot evaporate. It is water that evaporates, and therefore the total volume might be too low, and chloride concentration will be measured too high, because the sweat gets more concentrated in the course of evaporation.
But now, this explanation is in direct contrats to the fact, that the macr duct methods leads more often to QNS.
7.) Page 3, lines 102-103: The authors should provide a list of mutations that was included in the CFTR genetic analyses.
8.) The authors must state the day/month/year, when they accessed the CFTR2 database, since this database is dynamic.
Minor editing of english language necessary.
Author Response
Reviewer 3
Dear authors, I have read your manuscript with high interest. The direct comparison of these 2 methods could be helpful for centers and clinic to decide which test is more suitable to start with, and also to make a decision to change methods perhaps.
However, there are several points that need minor and sometimes major attention.
Re: We thank you for carefully evaluating our paper.
Major comments:
1.) The conclusion is merely a repition of results. In addition the authors state, that there is no differnce in the number valid samples. However, there are up to 25% more invalid tests with the macroduct systems. It is somehow unbelievale, that this is not significant.
Re: We have modified tables, discussion and conclusions according to the suggestions of the reviewers. However, the statistical differences are correct; in Table 2 we do not see the 25% difference on invalid tests reported by the reviewer. For example, regarding CFSPID subjects, with the Macroduct system we have 20/24 valid tests, while with Gibson-Cook 23/24 valid tests with p-value 0.25.
2.) The discussion should also be completely rewritten. The present text is not very structured, and it includes a lot of redundant information, that is already mentioned in the introduction.
Re: We apologize to the reviewer. We have rewritten the discussion.
3.) Table 3 is very hard to understand. It would be much better to rearange it, probably in the following way:
normal interm. altered pathologic
Gib.-Cooke 20 1 0
Macroduct 21 0 2
However, as you can see, I also do not understand your numbers. The number Macroduct and Gibson-Cooke do not match, and also the numbers between table 1 and table 2 do not match.
Re: Thank you for your comment, we have revised Table 3 and accepted your suggestion to make the table more understandable, and we also added the number of invalid tests in the tables to better understand the total number of tests performed. We also revised Table 2 by adding the total number of tests performed next to the percentages.
Minor comments:
4.) page 4, lines 157-161: This paragraph is quite unclear written. Please rewrite to make it more concise.
Re: Thanks for your suggestion, we have modified the sentence in the results section.
5.) page 4, line 166 - page 5 line 171: The data for SC should be presented in a graph. Additionally, the authors should definitely present the only significant difference. This should be presented in the same graph with different symbols/colours for NBS, CFSPID, transfused.
Re: Thanks for the suggestion, we have added a figure with the SC concentrations comparing the two collection methods in the three groups of subjects.
6.) Introduction, page 2 line 58: "The Gibson-Cooke method involves a significant risk of chloride evaporation ... "
First of all chloride from an aqueous solution cannot evaporate. It is water that evaporates, and therefore the total volume might be too low, and chloride concentration will be measured too high, because the sweat gets more concentrated in the course of evaporation.
But now, this explanation is in direct contrats to the fact, that the macr duct methods leads more often to QNS.
Re: Thank you for highlighting this point, you are absolutely right and we apologise. We have edited the sentence. As for the QNS with the Macroduct, we think the problem was both the complexity of drawing sweat from the Macroduct's coiled plastic tube with a syringe and the fact that the sweat collection surface area was smaller compared to the filter paper method as we reported in the discussion.
7.) Page 3, lines 102-103: The authors should provide a list of mutations that was included in the CFTR genetic analyses.
Re: Thanks for the suggestion, however, it is impossible to report all CF causing variants in the manuscript. We reported the CFTR2 link website and we add the date of uptade.
8.) The authors must state the day/month/year, when they accessed the CFTR2 database, since this database is dynamic.
Re: Thanks for highlighting this point; we added the date of CFTR2 database uptade.
Round 2
Reviewer 1 Report
I thank the authors for their responses. While I still believe that the novelty, sample size, and potential impact of the present study are lacking, I agree that some readers may interested due to the pediatric cohort of positive neonatal screens. I will defer to the editors of the journal on the ultimate decision on publication.
Several misspelled words are present in the revised manuscript. An additional spelling check should suffice.
Reviewer 3 Report
Dear authors,
thank you for the revision of your manuscript, which is (in my view) now much better understandable for the readers.
Dear editors,
there are a few sentences which might need some english editing. For example page 7, lines 246-250.
However, this could be possibly corrected during proof reading.